# Modeling the Contact Mechanics of Hydrogels

**Martin H. Müser [1],\* , Han Li [1] and Roland Bennewitz [2]**

[1]  Department of Materials Science and Engineering, Universität des Saarlandes,
    66123 Saarbrücken, Germany; s8haliii@stud.uni-saarland.de
[2]  INM—Leibniz Institute for New Materials and Physics Department, Saarland University, Campus D2 2,
    66123 Saarbrücken, Germany; Roland.Bennewitz@leibniz-inm.de
\*  Correspondence: martin.mueser@mx.uni-saarland.de

**Abstract:** A computationally lean model for the coarse-grained description of contact mechanics of hydrogels is proposed and characterized. It consists of a simple bead-spring model for the interaction within a chain, potentials describing the interaction between monomers and mold or confining walls, and a coarse-grained potential reflecting the solvent-mediated effective repulsion between non-bonded monomers. Moreover, crosslinking only takes place after the polymers have equilibrated in their mold. As such, the model is able to reflect the density, solvent quality, and the mold hydrophobicity that existed during the crosslinking of the polymers. Finally, such produced hydrogels are exposed to sinusoidal indenters. The simulations reveal a wavevector-dependent effective modulus $E^*(q)$ with the following properties: (i) stiffening under mechanical pressure, and a sensitivity of $E^*(q)$ on (ii) the degree of crosslinking at large wavelengths, (iii) the solvent quality, and (iv) the hydrophobicity of the mold in which the polymers were crosslinked. Finally, the simulations provide evidence that the elastic heterogeneity inherent to hydrogels can suffice to pin a compressed hydrogel to a microscopically frictionless wall that is undulated at a mesoscopic length scale. Although the model and simulations of this feasibility study are only two-dimensional, its generalization to three dimensions can be achieved in a straightforward fashion.

**Keywords:** hydrogels; contact mechanics; simulation; elastomers

## 1. Introduction

Hydrogels are random networks of cross-linked hydrophilic polymers. The strong attraction between polymer network and solvent leads to a swelling where the water content can exceed the polymer content by factors of ten and more. Hydrogels can be prepared as very soft materials with elastic moduli as low as a few hundred Pascals. Their mechanical properties have recently attracted great attention in a number of scientific research areas, in particular, in studies of hydrogels as biomaterials. Hydrogels resemble the extracellular matrix, and it has been shown that when using them as cell culture substrates, hydrogel stiffness is a key factor in cell spreading [1], cell motility [2], and even in the differentiation of stem cells [3]. Hydrogels also serve as materials for contact lenses and as such have been studied in their role as water-based lubricants [4–6].

In this paper, we propose a computationally effective coarse-grained model of hydrogels and demonstrate its potential by studying the scale-dependence of elastic properties and of contact mechanics. Relevant length scales in this problem include the persistence length of the polymer, the density of cross-links between polymers, the resulting mesh widths in the hydrogel, and possibly the characteristic decay length of inhomogeneity in the cross-link density. The scale dependence of elastic properties plays an important role in all of the applications introduced above. In the biophysics of cells, it is debated whether the elasticity of the cell culture substrate is important at the scale of

the cell diameter (10–30 μm) or at the molecular scale of the cell attachment [1,7,8]. In tribology, the scale-dependent Persson contact theory has led to a significant progress in the prediction of friction [9,10]. For hydrogels, the mesh size and the solvent viscosity have been identified as key parameters for friction [11]. Scaling concepts such as the relation between elastic modulus and mesh size $E \propto \zeta^3$ suggested by De Gennes [12] need to be refined for deformations at small length scales.

One of the standard biomaterial hydrogels is poly(ethylene glycol) diacrylate (PEGDA). Polymers of ethylene glycol repeat units are cross-linked by bonds between the acrylate groups at their two ends. We take structure, preparation, and mechanical properties of PEGDA hydrogels as an inspiration for our study. The elastic modulus of PEGDA hydrogels can be tuned by variation of the polymer density at the time of cross-linking, which is initiated by a chemical catalyst and UV light [13–15]. Cross-linking density and mesh size are then determined by the connectivity, i.e., the average number of polymers connected in one cross-link, and by the degree of entanglement [13]. The elastic modulus of hydrogels is measured macroscopically by oscillatory shear rheology, at the micrometer scale by indentation with colloidal probes [16,17], and on nanometer scale by atomic force microscopy (AFM) [8]. It has been found important to prepare hydrogels against a hydrophilic cover plate in order to avoid gradients of decreasing polymer content towards the surface of the hydrogel [18]. The elastic properties of hydrogels are often described by the classical concepts of rubber elasticity. Such description is appropriate for length scales larger than the relevant length scales of hydrogels introduced above and if the average distance between cross-links is large compared to the persistence length of the polymer, so that linear entropic elasticity dominates the interaction between crosslinks. The latter condition is not fulfilled for the random networks of structural fibers in the cell structure and, therefore, the modeling of random networks with semi-flexible polymers has received recent attention in biophysics [19,20]. The elastic response at small length scales, where the structure of the hydrogels comes into plays, has been addressed less often and is one of the key objectives in our study.

The approach, which is proposed here for the modeling of hydrogels, is centered at the mesoscale in that the used concepts are inspired from both atomistic descriptions and continuum modeling. As such one of our repeat units may represent anything between one or two chemical monomers and a single bead of a close-to-ideal Gaussian chain, in which case one repeat unit would represent several dozens or even hundreds of real monomers. Special emphasis is placed on the question how to generate hydrogels, such that their crosslink structure is not defined by "prejudice", for example, by eliminating randomly bonds in a given crystalline network. Instead, we attempt to produce polymer structures that properly reflect their random-walk characteristics in response to the solvent quality and the attraction to or repulsion from the walls of the mold in which the polymers are crosslinked. However, at the current, explorative stage, we restrict our attention to two-dimensional hydrogels. Moreover, we only consider thermal equilibrium and thus, zero-frequency contact mechanics. Including proper dynamics, e.g., by coupling the simulations to a lattice Boltzmann method [21] or to a dissipative particle dynamics [22], is challenging but feasible. However, both add-ons would slow down the simulations tremendously.

The remainder of this paper is organized into a models-and-method section, a results section, and final discussions and conclusions. As the models-and-method section is quite central to our study, its proper description may be relevant to mention at this point. It starts with a small overview over standard polymer models, which refers in particular to the characterization of single-polymer structure in a solvent, most notably in a Θ solvent. We then introduce our model for intramolecular interaction, which has a small "twist" compared to standard models in that angular potentials are implicit rather than explicit. This is meant to improve computational efficiency. Since the model is slightly non-standard, an in-depth analysis of important properties is presented from end-to-end radius characteristics to single-polymer force-extension curves. These properties are ultimately decisive for the hydrogel's mechanical response functions. Readers familiar with polymer physics can certainly skip the corresponding Sections 2.1 and 2.2. In the remaining model-and-methods section, monomer-wall potentials, intermolecular interactions, i.e., those acting between non-bonded monomers, and the

crosslinking procedure are described along with a discussion of how the simulations are set up and how they relate to contact mechanics. Worth mentioning at this point may be that we opted for a coarse-grained description of short-range repulsion in terms of a Flory–Huggins parameter rather than through short-range two-body potentials. The rationale is that the Flory–Huggins parameter can be deduced quite directly from experiment while the parametrization of repulsive interactions would require simulations to be run to identify the proper parameters. Moreover, and perhaps more importantly, two-body repulsion becomes soft at coarse scales, so that it no longer possesses its advantage of automatically disallowing bond-crossing. This is because two coarse-grained entities may have identical center of masses at a relatively small energetic cost. Thus, in future, three-dimensional simulations, where bond-crossing shall be avoided, bond-crossing must be disallowed explicitly anyway. In the results section, hydrogels are finally exposed to sinusoidal indenters, whereby effective contact moduli can be deduced, which turn out to depend on pressure and wavelength. Having this type of information allows theoreticians more easily to address more general contact geometries than that of force-indentation curves of parabolic indenters having different radii of curvature.

## 2. Model and Methods

As argued in the introduction, the model must reproduce various generic features of hydrogels including their production. While advanced ways to properly coarse-grain polymers exist [19,20,23–31], we opted for a model that is extremely simple, in particular with respect to the evaluation of forces, so that the model can be efficiently simulated with molecular dynamics. Moreover, we consider this work a feasibility study, in which the generic features of polymers are explored rather than the properties of a specific polymer even if we keep PEGDA polymers in mind. To remain computationally efficient, we therefore studied a two-dimensional rather than a three-dimensional hydrogel.

In the following, different aspects of the models are described in separate sections. Some of them are highlighted in Figure 1 to convey a first impression of various aspects related to the model. To provide readers outside polymer physics with the necessary background to rationalize the various aspects of the model, an in-depth description of the model is given, including a short section containing text-book [32] content on polymer physics relevant to this work. This polymer-physics-101 review section precedes those that are specific to our model. It allows various terms to be introduced, which are later used to rationalize the model and the results.

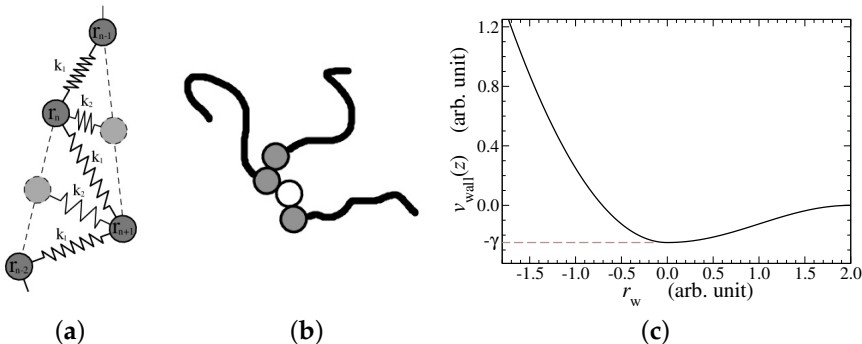

　　　　　(**a**)　　　　　　　　　　　(**b**)　　　　　　　　　　　(**c**)

**Figure 1.** Summary of various aspects of the used model. Schematic representation of (**a**) intra-molecular interactions and (**b**) crosslinking as well as (**c**) an implementation of the monomer wall interaction potential.

### 2.1. Standard Polymer Models

A simple model or representation of polymers is the freely jointed chain, in which the polymer is partitioned into $P_K$ Kuhn segments of fixed length $b$, which have no angle correlation within the chain. In the Kuhn model, the mean-square end-to-end radius is given by

$$\langle R_{EE}^2 \rangle = P_K\, b^2. \tag{1}$$

It applies to so-called $\theta$-solvents, in which $\langle R_{\mathrm{EE}}^2 \rangle$ scales linearly with the real degree of polymerization $P_{\mathrm{real}}$ for sufficiently long polymers. Note that a polymer is its own $\theta$-solvent. How $P_{\mathrm{K}}$ and $b$ are conventionally chosen is discussed further below.

The distribution function $\mathrm{Pr}(R_{\mathrm{EE}})$ resembles that of a Gaussian—except for very large values of $R_{\mathrm{EE}}$. Thus, the coarsest possible description of a polymer, in which (small) mechanical forces only apply to chain ends—as it happens when crosslinkers attach to them—is achieved by modeling the entire polymer with a single thermalized spring having a zero equilibrium length and a stiffness of

$$k_{\mathrm{EE}} = \frac{D\,k_B T}{R_{\mathrm{EE}}^2},$$
(2)

where $k_B T$ is the thermal energy and $D$ the spatial dimension. Thermal fluctuations need to be reflected, because they encapsulate the effect of the polymer's internal degrees of freedom. Alternatively, the chain could be partitioned into $P_{\mathrm{G}}$ thermal springs, each of which has a stiffness of $P_{\mathrm{G}} k_{\mathrm{EE}}$ to yield another good approximation to the true $R_{\mathrm{EE}}$ distribution function. This constitutes a Gaussian chain, whose configurations can be interpreted as realizations of random paths satisfying—on average—the diffusion equation. Partitioning the Gaussian chain into many small segments allows the (distribution of) polymer configurations to be represented when external potentials are present, e.g., in the form of an impenetrable wall [33].

As an incidental remark, it is noted that the freely jointed chain and the Gaussian chain show the same linear response regarding their force-extension characteristics. Differences in the two models only become apparent when the polymers are stretched by a significant fraction of $R_{\mathrm{EE}}$. The Gaussian chain does not change its stiffness, while the freely jointed chain becomes very stiff when the two ends are pulled $P_{\mathrm{K}} b$ apart. Of course, real force-extension curves lie in between those limits, see also Section 2.2.3.

A Kuhn segment may contain several repeat units that cannot be approximated as being freely jointed due to implicit or explicit bond angle potentials. If such potentials exist, the thermal expectation value $\langle \cos \vartheta \rangle$ can differ from zero, where $\cos \vartheta_n = \mathbf{a}_n \cdot \mathbf{a}_{n+1}/a_0^2$. Here $\mathbf{a}_n$ is the bond—or rather segment—vector $n$, which is numerated from 1 to $P_{\mathrm{r}}$, while $a_0$ refers to the fixed segment length and $P_{\mathrm{r}}$ to the number of repeat units into which a Kuhn segment is discretized. Note that the product $P_{\mathrm{r}} P_{\mathrm{K}}$ may but does not have to coincide with $P_{\mathrm{real}}$. In addition, the term "segment" refers to a fraction of the polymer, which may represent just one but, in most cases, is meant to encompass several (real) monomer units.

Assuming that the probability density for bond or segment angles satisfies $\mathrm{Pr}(\vartheta) = \mathrm{Pr}(-\vartheta)$ and that thermal fluctuations of different bond angles are independent from each other, it follows that

$$\left\langle \exp\left( i \sum_{n=n_1}^{n_1+\Delta n} \vartheta_n \right) \right\rangle = \langle \cos \vartheta \rangle^{\Delta n}$$
(3)

This in turn implies that the correlation of the chain's orientation decays according to

$$\langle \cos \vartheta_{\Delta n} \rangle = e^{-\Delta n\, a_0 / \lambda_{\mathrm{P}}}$$
(4)

with the so-called persistence length being

$$\lambda_{\mathrm{P}} = \frac{-a_0}{\ln \langle \cos \vartheta \rangle}.$$
(5)

If the chain length is much less than $\lambda_{\mathrm{P}}$, it behaves like a rigid rod, while chains are floppy when their lengths clearly exceed $\lambda_{\mathrm{P}}$.

Equation (3) also allows the end-to-end radius in the more refined model to be determined to

$$
\begin{aligned}
R_{\mathrm{EE}}^2 &= \sum_{m=1}^{P_{\mathrm{r}}} \mathbf{a}_m \cdot \sum_{n=1}^{P_{\mathrm{r}}} \mathbf{a}_n \\
&= a^2 \sum_{m=1}^{P_{\mathrm{r}}} \left( \sum_{n=1}^{m-1} c^{m-n} + 1 + \sum_{n=m+1}^{P} c^{n-m} \right) \\
&= a^2 \left\{ P_{\mathrm{r}} \frac{1+c}{1-c} - \frac{2\,c\,(1-c^{P_{\mathrm{r}}})}{(1-c)^2} \right\}
\end{aligned}
\tag{6}
$$

with $c = \langle \cos \vartheta \rangle$. In the limit $c^{P_{\mathrm{r}}} \to 1$, the chain resembles a stiff rod, in which case $R_{\mathrm{EE}} = P_{\mathrm{r}}\,a$, as can be deduced from L'Hôspital's rule. In the limit $c^{P_{\mathrm{r}}} \ll 1$, the chain resembles a Gaussian chain satisfying $R_{\mathrm{EE}} \propto \sqrt{P_{\mathrm{r}}}$. The Kuhn length $b$ is commonly chosen twice the persistence length, so that for long chains the number of Kuhn segments needs to be set to

$$
P_{\mathrm{K}} = \frac{P_{\mathrm{r}}}{4} \, \ln^2 \langle \cos \vartheta \rangle \, \frac{1 + \langle \cos \vartheta \rangle}{1 - \langle \cos \vartheta \rangle}.
\tag{7}
$$

### 2.2. Intramolecular Interactions

When modeling the mechanical properties of a hydrogel it is desirable to dispose of a description in which the representation of the polymer can be set arbitrarily somewhere between fine and coarse-grained scales. We attempt to achieve this with the following energy expression

$$
V_{\mathrm{intra}} = \frac{k_1}{2} \sum_{n=1}^{P} (r_{n-1,n} - a_0)^2 + \frac{k_2}{2} \sum_{n=1}^{P-1} \left( \mathbf{r}_n - \frac{\mathbf{r}_{n+1} + \mathbf{r}_{n-1}}{2} \right)^2,
\tag{8}
$$

where $P$ is the number of segments into which the "polymer path" is discretized and $\mathbf{r}_{n-1,n} = \mathbf{r}_n - \mathbf{r}_{n-1}$, which is identical to the segment vector $\mathbf{a}_n$. We transition from the letter $a$ to $r$, because the constraint of a fixed bond/segment length has been skipped. No claim of novelty is made regarding this expression, even if we could not identify literature in which precisely this expression was introduced. The common way to proceed is to introduce an explicit energy penalty on the bond angle rather than to guide repeat units implicitly towards the center-of-mass position of its nearest neighbor in the chain.

The rationale for the used model is the ease with which all terms appearing on the r.h.s. of Equation (8) can be coded and computed. Moreover, the model contains all above-mentioned coarse-grained models as limiting cases and its parameters are easily set to match target values for $a$ and $\langle \cos \vartheta \rangle$. For example, from Equation (8), it can be readily seen that $k_1 = \infty$, $k_2 = 0$, and $a_0$ being finite yields the freely jointed chain. In this limit, it would be most efficient to implement constraints rather than to use small time steps. In the opposite limit, i.e., when making $k_1$ finite and setting $a_0$ and $k_2$ to zero, a Gaussian chain is obtained, which might be useful for the modeling of a hydrogel in which extremely long polymers are crosslinked at their ends. Choosing a positive value of $k_2$ introduces a bending stiffness to the polymer, because this creates a force pushing a monomer towards the center-of-mass coordinates of its two neighbors, whereby the chain straightens out. In the following, the generic properties resulting from the energy expression in Equation (8) are explored.

### 2.2.1. Effective Segment Length

When inspecting Equation (8), it might be tempting to believe that the term proportional to $k_2$ only has a small effect on the root-means-square (rms) segment length $a_{\mathrm{rms}}$, which however, is incorrect unless $k_2$ is small compared to $\min(k_1, k_B T / a_0^2)$. The effect of $k_2$ on $a_{\mathrm{rms}}$ is most easily quantified in a one-dimensional model, because this constitutes a harmonic potential, which is amenable to analytical treatments. Towards this end, we explore the continuum approximation that results from expressing

the displacements of atoms $u_n \equiv x_n - x_{n,0}^{\text{eq}}$ from the their equilibrium positions $x_{n,0}^{\text{eq}} = n\,a_0$, in which case $u_n \to u(x)$ and $(u_n - u_{n-1}) \to a_0\,u'(x)$. As such, we may write

$$
\begin{aligned}
a_{\text{rms,1D}}^2 &= a_0^2 + \left\langle (x_n - x_{n-1})^2 \right\rangle \\
&\triangleq a_0^2 + \left\langle \{a_0\,u'(x)\}^2 \right\rangle .
\end{aligned}
\tag{9}
$$

The continuum approximation to the model in one spatial dimension reads

$$
V_{\text{pot}} = \frac{1}{2} \int dx \left[ \frac{k_1}{a_0} \{a_0 u'(x)\}^2 + \frac{k_2}{4\,a_0} \{a_0^2 u''(x)\}^2 \right].
\tag{10}
$$

In a Fourier representation, this expression allows a wavenumber dependent stiffness to be defined as

$$
\tilde{k}_{\text{eff}}(q) = k_1\,(a_0\,q)^2 + k_2\,(a_0\,q)^4/4.
\tag{11}
$$

Due to Parseval's theorem, mean squares of fields can be either added up in real space or in the Fourier representation. This is why we may write using equipartition

$$
\begin{aligned}
\left\langle \{a_0 u'(x)\}^2 \right\rangle &= \frac{k_B T}{k_1\,\pi} \int_0^\pi d\tilde{q}\,\frac{1}{1 + k_2 \tilde{q}^2/(4\,k_1)} \\
&= \frac{k_B T}{k_{\text{eff}}}
\end{aligned}
\tag{12}
$$

with $\tilde{q} = q\,a_0$ and

$$
k_{\text{eff}} = k_1\,\frac{\sqrt{\pi^2 k_2/(4\,k_1)}}{\operatorname{atan}\sqrt{\pi^2 k_2/(4\,k_1)}}.
\tag{13}
$$

Assuming continuum dispersion relationships up to $q_{\max} = \pi/a_0$ is certainly simplifying and using the correct ones would be feasible, at least when integrating the pertinent expression for $k_{\text{eff}}$ numerically. It would lead to a slightly reduced value of $k_{\text{eff}}$, because the continuum approximation overestimates the stiffness at the border of the Brillouin zone. However, it is felt that the current treatment is sufficiently refined at this explorative stage and that having simple analytical expressions is also advantageous.

An accurate analytical treatment of the model in two spatial dimensions is substantially more elaborate than in one dimension, because the potential is no longer a harmonic function of the displacements. The best we are currently able to do is to use $k_{\text{eff}}$ as an effective local bond stiffness. The thermal average of the second moment of the instantaneous segment length $r$—$a_{\text{rms}}^2$ being nothing but $\langle r^2 \rangle$ averaged in two dimensions rather than in one—then becomes ($\beta = 1/k_B T$, tildes indicate that a variable is expressed in units of $\sqrt{k_B T/k_{\text{eff}}}$):

$$
\tilde{a}_{\text{rms,2D}}^2 = \frac{\int d^2 r\, r^2\, e^{-\beta k_{\text{eff}}\,(r-a_0)^2/2}}{\int d^2 r\, e^{-\beta k_{\text{eff}}(r-a_0)^2/2}}
\tag{14}
$$

$$
= \frac{(2 + \tilde{a}_0^2)\,e^{-\tilde{a}_0^2/2} + \sqrt{\frac{\pi}{2}}\,\tilde{a}_0\,(3 + \tilde{a}_0^2)\left\{ \operatorname{erf}\left(\frac{\tilde{a}_0}{\sqrt{2}}\right) + 1 \right\}}{e^{-\tilde{a}_0^2/2} + \sqrt{\frac{\pi}{2}}\,\tilde{a}_0\left\{ \operatorname{erf}\left(\frac{\tilde{a}_0}{\sqrt{2}}\right) + 1 \right\}}
\tag{15}
$$

so that the two asymptotic regimes satisfy

$$
a_{\text{rms,2D}}^2 \approx \begin{cases} a_0^2 \cdot \left(1 + \frac{3\,k_B T}{k_{\text{eff}}\,a_0^2}\right) & \text{if } \tilde{a}_0 \gg 1 \text{ (as in Kuhn model)} \\[2mm] \frac{2\,k_B T}{k_{\text{eff}}} \left(1 + \sqrt{\frac{\pi\,k_{\text{eff}}}{8\,k_B T}}\right) & \text{if } \tilde{a}_0 \ll 1 \text{ (as for Gaussian chains)} \end{cases}.
\tag{16}
$$

When $k_2 a_0^2$ is large compared to $k_B T$, it is clear that the chain is locally quasi-one dimensional and that the one-dimensional formula for $a_{\text{rms}}^2$, see Equation (12), is the appropriate choice, while in the opposite case, the two-dimensional treatment resulting in Equation (15) is to be preferred. A switching function $s$ is needed to embed both limits in one formula, e.g., through

$$a_{\text{rms}}^2 \approx s\, a_{\text{rms,2D}}^2 + (1 - s)\, a_{\text{rms,1D}}^2 \tag{17}$$

At this point, we are only in a position to argue heuristically. Assuming the switching function to be exponential in $k_2$, the only functional form it may have from a dimensional analysis point of view is

$$s = e^{-\beta g\, k_2\, a_0^2}. \tag{18}$$

It was found that values of $g$ less to but in the order of unity provide satisfactory cross-over functions. This is demonstrated in Figure 2, which considers one model in which $k_1 = k_2$ is chosen and another one with $k_1$ being fixed at a value clearly exceeding $k_B T / a_0^2$.

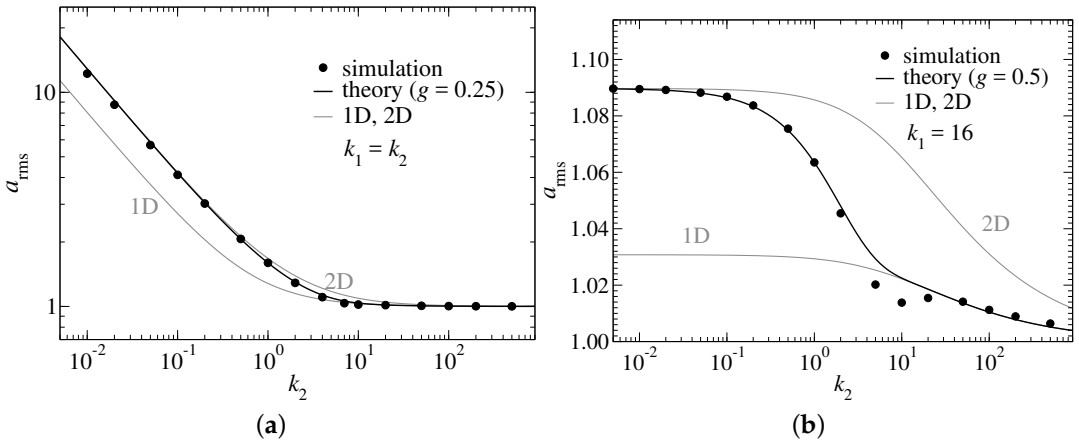

<div align="center">(<b>a</b>)　　　　　　　　　　　　　　　　　　　(<b>b</b>)</div>

**Figure 2.** Root-mean-square segment length $a_{\text{rms}}$ as a function of the spring stiffness $k_2$: (**a**) $k_1 = k_2$, (**b**) $k_1 = 16$. Spring stiffnesses are stated in units of $k_B T / a_0^2$ and segment lengths in multiples of $a_0$. Simulation data is shown as circles, theoretical results with black lines. The asymptotic behavior in the one-dimensional (1D) and two-dimensional (2D) case are shown with gray lines.

The scaling dependence of $a_{\text{rms}}$ differs in the two graphs shown in Figure 2, because the chain becomes similar but not identical to an ideal, two-dimensional Gaussian chain in the limit $k_1 = k_2 \to 0$. However, the segment length is approximately constant—with merely minor thermal increments compared to $a_0$—when $k_1$ is fixed such that it clearly exceeds $k_B T / a_0^2$. At large values of $k_2$ both models behave asymptotically as effectively one-dimensional chains. The small errors in Figure 2a between the predicted two-dimensional asymptotic prediction and the simulation data result from the chain not becoming a true Gaussian chain—due to the somewhat artificial choice of keeping $k_2$ identical to $k_1$.

2.2.2. End-to-End Radius

Obtaining analytical estimates for $R_{\text{EE}}^2$ requires approximations for $\langle \cos \vartheta \rangle$ to be known in addition to $a_{\text{rms}}$. Towards this end, the potential energy is rewritten in terms of the segment vectors $\mathbf{a}_n$ as

$$
\begin{aligned}
V_{\text{intra}} &= \sum_{n=1}^{P} \frac{k_1}{2} (a_n - a_0)^2 + \sum_{n=2}^{P} \frac{k_2}{8} (\mathbf{a}_n - \mathbf{a}_{n-1})^2 \tag{19} \\
&\approx \sum_{n=1}^{P} \frac{k_1}{2} (a_n - a_0)^2 + \sum_{n=2}^{P} \frac{k_2\, a_{\text{rms}}^2}{4} (1 - \cos \vartheta_n), \tag{20}
\end{aligned}
$$

The model resulting from the approximation in Equation (20) – in the limit $k_1 a_0^2/(k_B T) \to \infty$ is also known as a semiflexible chain. $\langle \cos \vartheta \rangle$ can be calculated from

$$
\begin{aligned}
\langle \cos \vartheta \rangle &= \frac{\int_0^{2\pi} d\vartheta \cos \vartheta\, e^{\tilde{k}_2 \cos \vartheta}}{\int_0^{2\pi} d\vartheta\, e^{\tilde{k}_2 \cos \vartheta}} \\
&= \frac{I_1(\tilde{k}_2)}{I_0(\tilde{k}_2)},
\end{aligned}
\tag{21}
$$

with $\tilde{k}_2 = \beta k_2 a_{\mathrm{rms}}^2/4$ and $I_n(...)$ being the modified Bessel function of the first kind.

Rather than reporting $\langle \cos \vartheta \rangle$, Figure 3 shows the dependence of $R_{\mathrm{EE}}$ on $k_2$ for the cases $k_1 = k_2$ and $k_1 a_0^2/(k_B T) \gg 1$. Analytical results are obtained by using our approximations for $a_{\mathrm{rms}}$, as shown in Figure 2, in Equation (6). The resulting predictions turn out quite satisfactory except for the case $k_1 = k_2 \ll k_B T/a_0^2$. However, when coarse graining polymers in a systematic fashion, $k_2$ will always disappear more quickly than $k_1$, since the $k_2$ term scales with the inverse fourth power of the wavelength, while the $k_1$ term only scales with the inverse square, as can be deduced, for example, from Equation (11). This would make our model approach the limit of an ideal Gaussian chain quite quickly so that analytical expressions for $R_{\mathrm{EE}}$ would be quasi exact. Remember that $R_{\mathrm{EE}}$ exceeds $a_0$ times the number of segments in that limit.

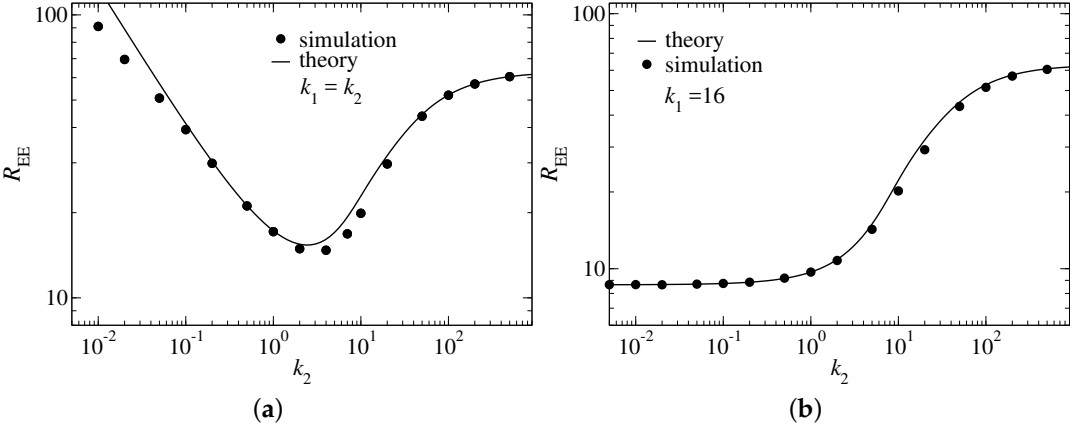

**Figure 3.** End-to-end radius $R_{\mathrm{EE}}$ as a function of the spring stiffness $k_2$: (**a**) $k_1 = k_2$, (**b**) $k_1 = 16$. Spring stiffnesses are stated in units of $k_B T/a_0^2$ and segment lengths in multiples of $a_0$. The number of segments is $P = 63$. Simulation data is shown as circles, theoretical results with black lines.

### 2.2.3. Force Extension Curves

An important property of polymers and the models describing them is that they are initially stretched easily while extended polymers are very stiff. Specifically, the effective (entropic) spring constant at small extensions is given by $k_{\mathrm{EE}}$—see Equation (2)—while the stiff covalent bonds become dominant at large extensions. The latter are reflected in our model through the harmonic term proportional to $k_1$.

For polymers with $P = 63$ (segments) and values of $k_2$ sufficiently small for the chains to be floppy, our model reproduces these scaling laws. Entropic springs act up to an extension of $x \gtrsim R_{\mathrm{EE}}$, while for large extensions, the linear $F = k_1 x/P$ scaling is obeyed quite closely. These claims are substantiated in Figure 4.

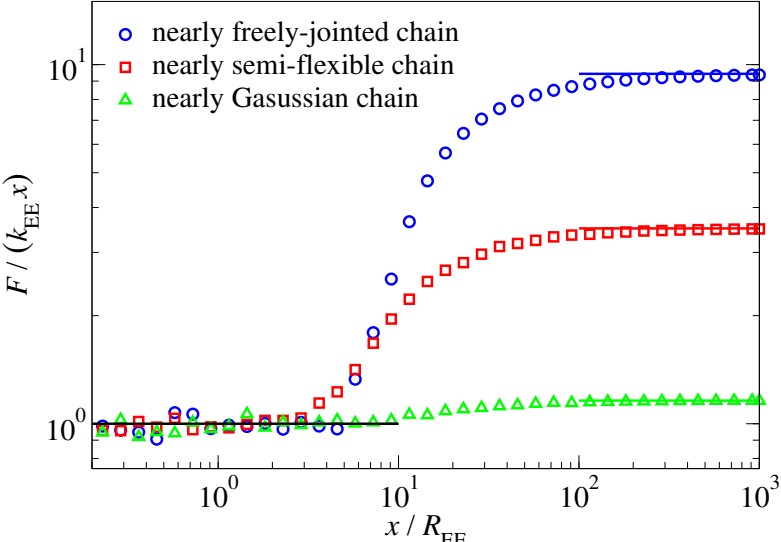

**Figure 4.** Extension force $F$ as a function of a polymer's extension $x$. $F$ is the mean force needed to keep one polymer end fixed at position $(x, 0)$ when the other end is fixed in the origin. It is normalized to $k_{EE} x$, where $k_{EE}$ is defined in Equation (2). The extension $x$ is normalized to $R_{EE}$. Symbols show simulation data. Parameters used: $P = 63$ (segments) for all data. In addition—$\tilde{k}_n$ defined here as $k_n a_0^2 / (k_B T)$— nearly freely jointed chain (blue circles): $\tilde{k}_1 = 16$, $\tilde{k}_2 = 0$, nearly semi-flexible chain (red squares): $\tilde{k}_1 = 4$, $\tilde{k}_2 = 4$, nearly Gaussian chain (green triangles): $\tilde{k}_1 = 1/16$, $\tilde{k}_2 = 0$. Colored lines represent the asymptotic predictions $F = k_1 x / P$ for stretched chains.

### 2.3. Monomer–Wall Interactions

In coarse-scale descriptions, the interactions between monomers and confining walls is commonly represented in terms of hard-wall potentials. While implementing such constraints is possible within molecular dynamics, we chose another route, namely, a finite but short-range repulsion between monomers and wall. We make the interaction as stiff as possible without having to reduce the time step by setting the curvature of the monomer-wall interaction potential similar in magnitude as the chain stiffness, specifically,

$$V_{\text{wall}-\text{rep}} = \frac{\max(k_1, k_2)}{2} \, \Theta(-r_{\text{p}}) r_{\text{p}}^2, \tag{22}$$

where $\Theta(..)$ is the Heavyside step function, while $r_{\text{p}}$ is the (signed) distance of a monomer from the wall. A negative sign of $r_{\text{p}}$ indicates a penetration into the wall. The rms-slope of the used wall profiles is always sufficiently small to have a unique value for $r_{\text{p}}$.

When mimicking walls attracting polymers more strongly than solvent, adhesion is added. To make it computationally tractable, it is chosen as

$$V_{\text{wall}-\text{adh}} = -\gamma \, \Theta(-r_{\text{p}}) - \frac{\gamma}{2} \left\{ 1 + \cos \left( 2 \pi \, \frac{r_{\text{p}}}{r_{\text{max}}} \right) \right\} \Theta(r_{\text{p}}) \, \Theta(r_{\text{max}} - r_{\text{p}}). \tag{23}$$

Here $\gamma$ is the surface energy and $r_{\text{max}}$ the range of interaction.

### 2.4. Intermolecular Interactions

When a polymer is not in its $\Theta$-solvent or when confining walls or other external potentials are present, it is necessary to include interactions between the polymer's repeat units beyond nearest neighbors, even when the polymer is described at a coarse scale. These interactions as well as those between monomers belonging to different polymers are lumped together as intermolecular interactions.

The leading-order approach in a density-functional or self-consistent field theory (SCFT) [25] to these interactions is to make an extra monomer "pay" an energy density of $\chi \cdot \rho(\mathbf{r})$ if positioned at a site $\mathbf{r}$, where $\rho(\mathbf{r})$ denotes the (mean) density. $\chi$ is also called the Flory–Huggins parameter from

the Flory–Huggins theory for solutions of polymers [34]. As such, $\chi$ is an indirect measure for the chemical potential of the solvent, i.e., water, at the given temperature. Our model thus, implicitly assumes the solvent to be able to enter or to leave the hydrogel so that the pressure exerted on the mold or confining wall can be interpreted as an osmotic pressure of the polymers.

A similar approach, as those pursued in SCFT is adopted here, however, with two approximations to make the approach computationally efficient. First, the density of a monomer is smeared out so that an estimate for the coarse-grained density is obtained

$$\rho_{\text{cg}}(\mathbf{r}) = \frac{1}{\sqrt{2\,\pi\Delta a^2}} \sum_n e^{-(\mathbf{r}-\mathbf{r}_n)^2/(2\,\Delta a^2)} \delta(\mathbf{r} - \mathbf{r}_n), \tag{24}$$

where $\Delta a$ has the unit length and $\mathbf{r}_n$ is the coordinate of monomer $n$. $\Delta a$ was set such that it was similar to $a_{\text{rms}}^2$.

The second approximation is that the force acting on monomer $n$ is not taken to result from the derivative of $\chi \int d^2r \rho_{\text{cg}}^2(\mathbf{r})/2$ with respect to $\mathbf{r}_n$ but this term is approximated with the gradient at a monomer's center-of-mass position, i.e., through

$$\mathbf{F}_n = -\chi \nabla \rho_{\text{cg}}(\mathbf{r})_{|\mathbf{r}=\mathbf{r}_n}. \tag{25}$$

Strictly speaking, this part of the model constitutes a force field rather than a potential. However, we expect differences between an "exact" treatment and the pursued approach to be small. Moreover, the main goal of the model is to obtain a scaling similar to that predicted by Flory, i.e., $R_{\text{EE}} \propto P^{3/(D+2)}$ in the limit of large $P$ when a single polymer is immersed in a good solvent. This scaling (or more precisely, the one with the exact rather than the Flory exponents [12,32]) will most certainly be achieved in our model, as it penalizes energetically a recurrent random walk. When parametrizing the model to a specific polymer-solvent system $\chi$ needs to be set such the correct prefactor is obtained.

Finally, the smearing out of density proposed in Equation (24) was done in an approximate fashion. First, a grid was used to which the effective monomer density $\rho(\mathbf{r}) = \sum_n \delta(\mathbf{r} - \mathbf{r}_n)$ was binned such that the center of mass remained unchanged. Rather than using a fast Fourier transform, the density at a given grid point was smeared out such that one ninth of it was reassigned to the original grid point as well as to its nearest, and next-nearest neighbors on a square lattice. The procedure is repeated four times, resulting in a smearing out that starts to look Gaussian. Density gradients were then obtained numerically on the grid and extrapolated linearly between grid points. Better approximations can certainly be achieved to the entire procedure by using finer grids and more coarsening iterations. However, for the choice of parameters used in the results section, extraordinarily large statistics are needed to detect any effects on the final observables of interest. Also the resulting coarse-grained densities, which are shown in Section 2.6, appear to look reasonable. Again, the two central aspects of our model are computational efficiency and the possibility to correctly set prefactors to the $R_{\text{EE}}(P)$ asymptotic scaling relations, both of which the used model can certainly achieve despite unconventional coarse-graining procedures.

Figure 5 visualizes the effects that intramolecular interactions have on the density profiles near a wall. In $\Theta$-solvent conditions, the effectively hard-wall boundaries lead to a depletion of monomer density near the walls up to a distance of order $R_{\text{EE}}$. The inter-molecular repulsion reduces the size of the depletion zone. Finally, adhesion induces an enhanced density nearby the walls.

Note that our model in its current formulation and parametrization does not provide significant impediments to bond-crossings. As such, it cannot be used in situations where the crosslink and polymer density are high and realistic dynamics are of interest in addition to thermodynamics. However, the common theoretical approaches to the elastic properties of hydrogels do not deem non-crossing domains as significant. More importantly, when designing a two-dimensional model with the intent to emulate a projection of a three-dimensional system, it is meaningful to allow for bond-crossing, because the projection of bonds may certainly cross.

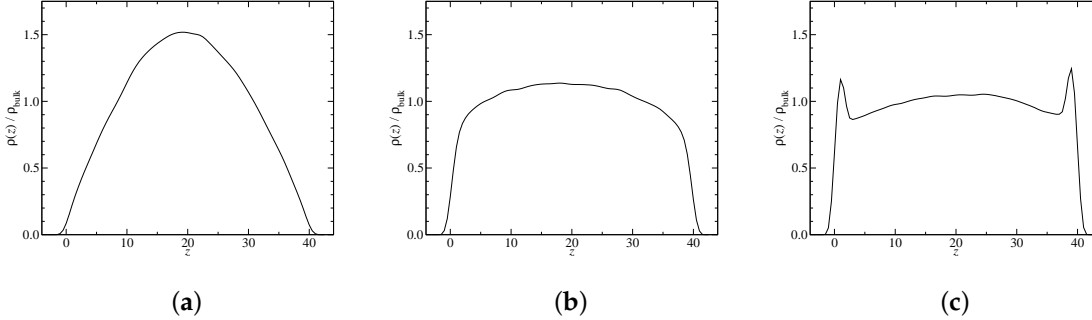

**Figure 5.** Monomer density profiles in a confined geometry. (**a**) Repulsive monomer-wall interactions and Θ solvent conditions, (**b**) good rather than Θ solvent, and (**c**) weak adhesion added. Values for the parameters are chosen as in the default model investigated in the results section. The two flat walls are positioned at $z = 0$ and $z = 40$. The density is normalized to the bulk density.

## 2.5. Crosslinkers

In order for a $D$-dimensional network of Gaussian chains to have a finite shear stiffness, a percolating network needs to be established in which $D + 1$ chains are connected to individual crosslinkers [12]. To emulate such networks, crosslink molecules with four beads were introduced. Each bead can connect to a maximum of one end-monomer of a polymer, as depicted schematically in Figure 1b. A connection is modeled with a harmonic spring of zero equilibrium length and a stiffness $k_1$, where $k_1$ pertains to the value used for the polymers.

As the crosslinkers were short molecules, no coarse-grained density was defined for them, neither were they coupled to the polymer's coarse-grained density. Connections between crosslinkers and polymer ends were established irreversibly after equilibration had taken place whenever a free crosslinking monomer approached an unconnected polymer end within $a_0/3$.

In our model the formation of crosslinks and crosslink topology evolves such that polymer ends may end up dangling. There is also the possibility of clusters to be formed that do not belong to the percolating cluster. As these constituents may also exist in reality, it was decided to not eliminate dangling ends or disconnected polymers. In real systems they certainly also affect the viscoelastic response functions until they are squeezed out of the hydrogel.

## 2.6. Model Parameters for Indentation Simulations

For the remaining part of this paper, the following parameters were selected to be: $P = 24$, $a_0 = 1.5$, $k_1 = 16$, $k_2 = 4$, and $\chi = 0.25$. This leads to a persistence length $\lambda_p$ of roughly 16 segments so that crosslinks were separated by polymers being approximately 1.5 times $\lambda_p$. The crosslinkers were modeled with $P = 4$, $a_0 = 1.5$, $k_1 = 10$, $k_2 = 10$, and $\chi = 0$. Finally, the temperature was set to $T = 0.25$.

The total number density of the monomers was set to unity, which resulted in a relatively dense mesh, which allowed us to test if the network would swell after releasing it from the mold, through the removal of the confining wall, and immersing it in water, by keeping $\chi$ unchanged. The density of crosslinkers, $\rho_{cl}$, states the number of crosslink monomers over the number of polymer ends. Four different values were considered $\rho_{cl} = 0$, 0.25, 0.83, and 1.67. Only $\rho_{cl} = 0.83$ and $\rho_{cl} = 1.67$ lead to the production of percolating networks.

There was no particular motivation for these parameters, except for $a_0$ roughly reflecting the C–C bond length when Å is taken as the unit of length. All other parameters were determined by a trial and error procedure until the produced networks were pleasing to the eye. It had been attempted to produce locally smooth polymer configurations and moderate density fluctuations in the network density. This is why all quantities are simply stated as numbers. The reader may, however, interpret the temperature of $T = 0.25$ as room temperature, $a_0$ as a C–C bond length, and rescale all data in

that unit system. A systematic coarse-graining of real polymers in three dimensions is deferred to a future study.

Finally, the box geometry in the results section was chosen to be 200-length units long and crosslinking took place in a "mold", which was $h_0 = 80$ height units high. When exposed to an indenter at the top wall, the perfectly flat bottom layer was kept unaltered. It thereby constitutes a boundary condition of zero displacement in normal direction while providing zero shear stress. Since (shear) stress components decay roughly proportional to $\exp(-qz)$ towards the bulk, $z$ being the normal distance from the (top) wall and $q$ the in-plane wavenumber, our system is essentially very large even with respect to the long-wavelength undulations.

## 2.7. Contact Mechanics

The usual approaches to contact mechanics are based on the assumption that knowing the (normal) surface displacement $u(\mathbf{r})$ of an elastic body, which is imposed through a frictionless contact with a counterbody, is sufficient to deduce the deformed body's elastic energy. In other words, $U_{\mathrm{ela}}$ is a functional of $u(\mathbf{r})$. For flat, linearly elastic bodies, this functional is most easily expressed in a Fourier representation

$$U_{\mathrm{ela}}[u(\mathbf{r})] = \sum_{\mathbf{q}} \frac{q\,E^*}{4}\,|\tilde{u}(q)|^2\,, \tag{26}$$

where $E^*$ is the contact modulus and $\mathbf{q}$ a wavevector, or, in our case of a 1+1 dimensional hydrogel, a wavenumber. The validity of Equation (26) furthermore assumes isotropy, and the surfaces to be extended, or, more precisely, to be periodically repeated. More so, conventional linear elasticity is a scale-free theory.

Equation (26) turns quickly into a poor approximation for soft elastomers. The main reasons for its breakdown are that elastomers in general and hydrogels in particular are spatially heterogeneous and their elasticity is scale-dependent, unless for ondulations whose wavelengths clearly exceed the characteristic spacing between crosslink points. Even worse, the elasticity of soft matter quickly deviates from linear elasticity. In order to nevertheless pursue theoretical approaches to contact mechanics, it remains to be understood how Equation (26) needs to be generalized to describe the contact mechanics of hydrogels.

The first generalization is the use of a scale-dependent elastic modulus, in which case the following substitution needs to be made

$$E^* \rightarrow E^*(q). \tag{27}$$

It allows two things to be achieved: First, the scale-dependent stiffness of (bulk) elastomers can be encoded—at least in a disorder-averaged sense, as shown, for example, in Section 3. Second, spatial heterogeneity in the direction normal to the interface can be reflected through the substitution made in Equation (27). The latter is well known from linear elasticity, where, for example, correction factors $f_{\mathrm{bc}}(q\,t)$ encode the finite thickness $t$ of the deformed body and the boundary conditions (bc) at the surface opposite to the interface, where bc may stand either for zero traction or zero (tangential) displacement. In addition, for deformations that are not frictionless at the primary interface, the surface displacement must be represented as a vector and $E^*(q)$ as a tensor of rank two.

$E^*(q)$ of a crosslinked network is expected to have no significant wavelength dependence on $q$, once $1/q$ clearly exceeds the distances on which the hydrogel is heterogeneous, say, a few times the typical distance between crosslinks. More generally speaking, $1/q$ needs to exceed any characteristic length in the system, such as $\lambda_{\mathrm{p}}$, however, it must not exceed the hydrogel's thickness, in which case $E^*(q)$ is sensitive to the boundary condition—e.g., constant stress versus constant displacement—on the opposite surface. In a non-crosslinked melt $E^*(q \rightarrow 0) \rightarrow 0$, since the (static) contact modulus of a melt is zero in the continuum limit.

*2.8. Numerical Methods*

Molecular dynamics (MD) was used to obtain the results presented in this study with a house-written C++ code, which was specifically designed for this study. In our MD simulations, the end points of the segments are treated as dynamical variables rather than the segments themselves. These end point are called monomers, even if they may encompass several real monomers. The Newton's equation of motion, which result from the interaction potential and from assigning a mass $m$ to each monomer, were integrated in time using velocity Verlet. Time steps were set to $\pi \sqrt{(k_1 + 2k_2)/m}/20$. Periodic boundary conditions were employed in the direction normal to the interface. Wall positions were held fixed.

In addition to the conservative forces acting on the monomers, whose enumeration we start at $n = 0$ rather than at $n = 1$ as for the bonds or segments. Constant temperature needs to be imposed, which was achieved with thermostats. An ideal thermostat emulates the (dynamical) effects of those degrees of freedom (DOFs) that have been coarse-grained out on the remaining DOFs. Here, however, we are predominantly interested in (static) thermodynamic properties, which is why a quick equilibration is desired. Towards this end, we use a Langevin thermostat, which consists of a damping term $-m\,\gamma\,\mathbf{v}_n$, where $\mathbf{v}_n$ is the velocity of monomer $n$ and a random force with mean zero, and a second moment of $2\,m\,\gamma\,k_B T/\Delta t$, where $\Delta t$ is the time step. The damping constant is best set such that the slowest internal mode in a polymer, namely $R_{EE}$, is close to being underdamped, which can be reasonably well achieved by choosing $\gamma = \sqrt{k_{\text{eff}}/(P\,m)}$.

## 3. Results

*3.1. Analysis Procedure and Non-Affine Deformations*

Continuum mechanics assumes affine deformations, i.e., the microscopic displacement field is identical to that of the macroscopically strained body. Thus, a small, affine deformation that is imposed on a surface leads to a response in the solid with a strain or displacement field having only that in-plane wave vector. Disordered systems in general and elastomers in particular are good examples of systems undergoing non-affine deformations when being strained or stressed. This non-affinity is a natural consequence of the structural and thus elastic heterogeneities that exist at small scales.

As can be appreciated in Figure 6, the studied in-silico hydrogel also undergoes non-affine deformations: the spatially resolved stress profile, which results from the indentation of an indenter with a single non-zero wavevector roughness, clearly deviates from a single sine wave. Moreover, the stress profiles averaged over two subsequent runs ressemble each other, although each run was averaged over several hundred times the (apparent) relaxation times deduced from the stress auto-correlation function. The correlation of two subsequent runs is therefore not the consequence of poor sampling but primarily of elastic anisotropy.

Non cross-linked polymers also reveal stress profile with large local variation when averaged only over a few $O(10^4)$ time steps. However, the curves turn into smooth periodic functions after averaging over a few $O(10^5)$ time steps for the given period and load. Inspection of the hydrogel's microscopic configuration reveals that stress spikes, i.e., points carrying much load, correlate with crosslinkers being close to the interface. These points barely move in time when the sample is crosslinked. The elastic solid is thus, pinned with respect to the undulation. Following previous arguments on the interaction between disordered solids and period surface potentials, the free-energy barrier preventing a thermally induced jump by one period is expected to grow with the square root of the contact size, although this argument only applies to a manifold of fixed height; fixed shape would require more elaborate considerations. Thus, for large contact sizes but thin slabs, the elastic manifold would be pinned on astronomically large times, even though the walls are perfectly smooth microscopically.

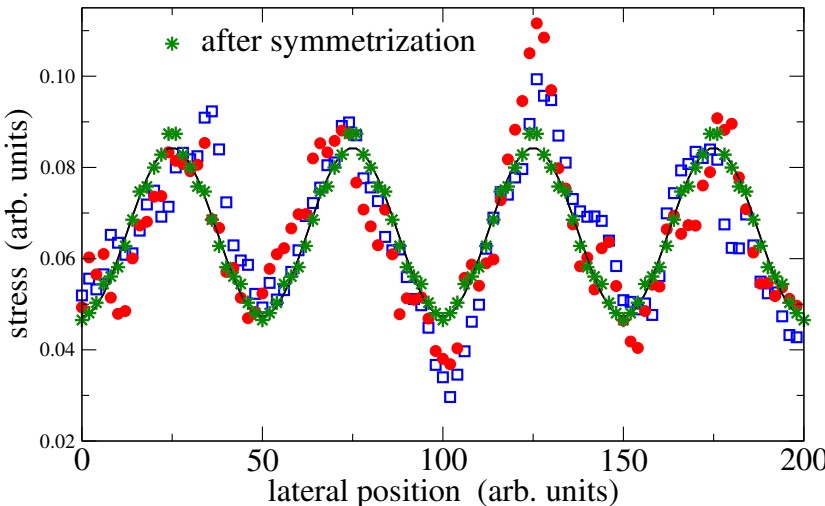

**Figure 6.** Spatially resolved stress at a surface with a sinusoidal indenter. Red circles and blue squares show averages taken over $4 \times 10^5$ time steps from two subsequent simulation runs, in which the first $10^5$ time steps are discarded for equilibration. Green stars show the average over the two curves after applying the allowed symmetry operations. The black line is a fit to the resulting curve with the function $\sigma(x) = \sigma_0 - \sigma_1 \cos(q\,x)$.

From the discussion in the last paragraph, it might be tempting to conclude that the relation $\tilde{\sigma}(q) = q\,E^*(q)\,\tilde{u}(q)/2$, is of rather limited benefit with respect to modeling the contact mechanics of hydrogels—even when the imposed strains and stresses are small. However, the relation may still hold on average, in which case continuum mechanics could still predict mean values. Averaging can be done either by averaging over different hydrogels realizations (running several production processes of the hydrogel with different random seeds), or, by exploiting symmetry when the simulation cell contains sufficiently many periods. For the data shown in Figure 6, there are four translations, each with or without mirror reflection, resulting in an effective average over eight data sets per period. The such obtained data is clearly dominated by a single sine wave. Note that the results of the fitting procedure do not depend upon whether the original data or the symmetrized data is fitted.

Thus, statistically speaking, the deformations are still affine. Of course, when the strain amplitudes start being large, linear elasticity is no longer sufficient for the description of the interfacial stress profiles. Higher-harmonics must then be included for a faithful description of the experimental or simulation data.

### 3.2. Effect of Load and Solvent Quality on the Effective Contact Modulus

Essentially all solids stiffen when being compressed. The reason for simple, i.e., densely packed solids is that atoms are pushed more deeply into the repulsive parts of their potentials, where the potentials' curvature—and thus, effective spring constants—increase. In the case of hydrogels, compression leads to an increase in the crosslink density, which is argued to linearly affect a hydrogel's shear stiffness at small crosslink densities [12]. To explore these effects in the pursued model, the mold is replaced with an adhesionless indenter having a sinusoidal shape at fixed thickness or height $h_0$, i.e.,

$$h(x) = h_0 + \tilde{h}_q \cos(q\,x). \tag{28}$$

The value of $\tilde{h}_q$ is set such that $q\,\tilde{h}_q = 1/4$. By choosing $q\,\tilde{h}_q$ constant, it is ensured that the contact line—as it would be determined in a continuum approximation—is constant, which facilitates the deduction of meaningful $E^*(q)$ relations. A relatively large value of $q\,\tilde{h}_q$ guarantees relatively good signal to noise ratios.

In Figure 7, the stress profiles emerging in response to a sinusoidal indenter with fixed surface shape—specifically an indenter having a wavelength of 100 as the one depicted in Figure 8—is shown

for (i) two different concentrations of crosslinkers and (ii) two different heights. The responses of the two different $\rho_{cl}$ values at the original height of $h = h_0$ are compared first for an undulation having a wavelength of 100. The crosslinked sample ($\rho_{cl} = 0.83$), while exerting less pressure on average on the confining wall than the freely floating polymers ($\rho_{cl} = 0$), reveals the slightly larger stress oscillations. Both is readily explained: Crosslinking reduces the effective repulsion between previously disconnected monomers, which in turn diminishes the force exerted on a confining wall. On the other hand, the larger undulation in the connected network's pressure results from the network being a solid rather than a fluid.

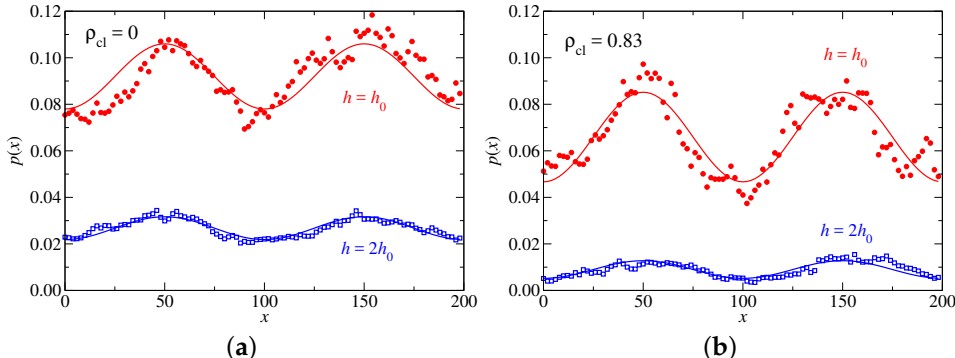

**Figure 7.** Pressure profile $p(x)$ for two different crosslinker densities (**a**) $\rho_{cl} = 0$ and (**b**) $\rho_{cl} = 0.83$ at two different heights $h_0$ and $2h_0$, where $h_0$ is the height of the mold in which the polymers were crosslinked.

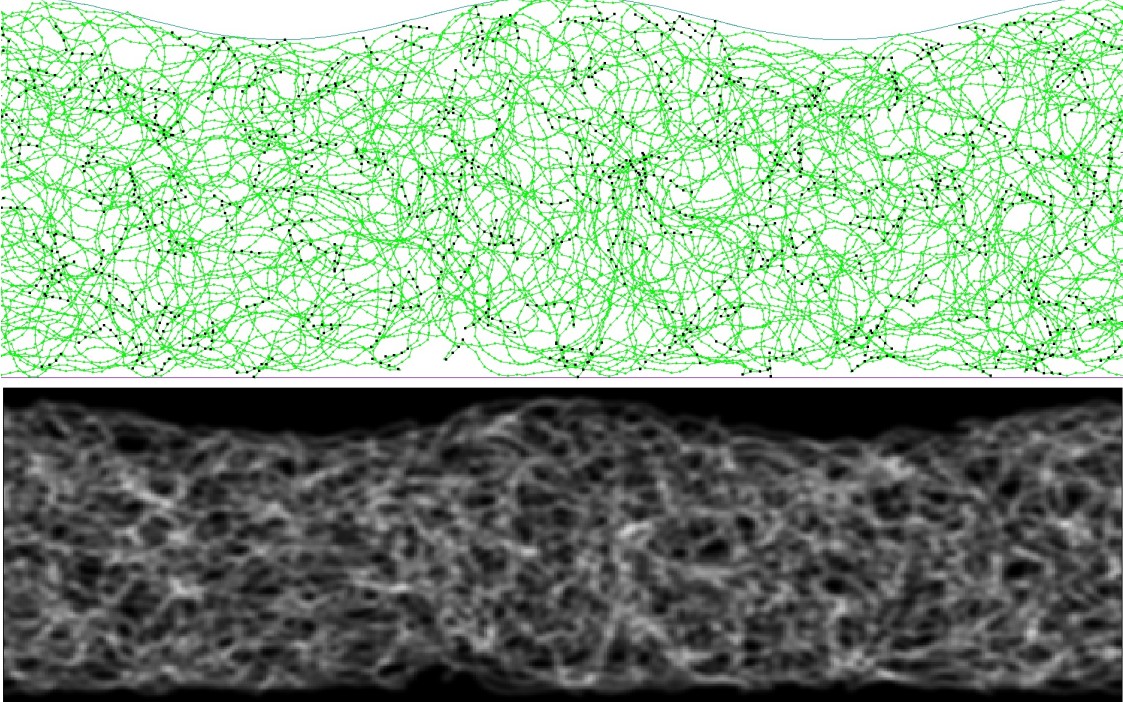

**Figure 8.** (**Top**) Representative configuration. Regular polymers are represented in green and crosslinking molecules in black. The positions of the two confining walls are indicated with thin, straight lines. The mean distance between the two confining walls is 64 and the length of the simulation is 200 in reduced units. (**Bottom**) Coarse-grained density associated with the configuration shown on top. Black and white represent zero and maximum density respectively.

The non-isotropic stress at an indenter-fluid interface revealed in Figure 7a might appear counterintuitive from a continuum perspective, because non-isotropic pressures imply a system's ability to sustain non-zero shear. However, it needs to be kept in mind that any system having structure

at a lengthscale $\lambda$ in one way or another is not isotropic, which is why it can sustain moderate amount of shear imposed at that or smaller lengthscales. In our specific case, stiff chains resist being squeezed into an undulation with a wavelength being less than the polymer's persistence length—whether or not it belongs to a crosslinked network. A very small, non-isotropic stress would have even resulted for a freely jointed polymer model, in which case $R_{EE}$ would have been the characteristic lengthscale below which it reveals noticeable "solid-like" behavior. The perhaps most simple argument to sustain this latter claim is that long random paths cannot be squeezed into narrow structures or ondulations, while short random paths would not object.

We will now consider the swelling of the hydrogel, i.e., its expansion upon reducing the confinement by moving the undulated indenter to a position allowing the hydrogel to double its mean original width. Solvent is implicitly supplied in an unlimited fashion by keeping $\chi$ unchanged. Normal pressures in both samples become distinctly smaller. The reduction is noticeably larger for the connected hydrogel (almost a factor of 10) than for the polymers floating freely in solution (slightly less than a factor of 3). Also this is easily rationalized: effective repulsion between disconnected polymers are relatively long ranged, while the hydrogel is a network with a finite equilibrium height. Thus, in the latter case, only the few chains that do not participate in the percolating hydrogel would exert a small osmotic pressure on the confining wall once the width between substrate and indenter exceeds the hydrogel's equilibrium width.

Although the amplitude of the imposed surface undulation is identical for the $h = h_0$ and the $h = 2\,h_0$ simulations, the pressure variations are much reduced after the expansion to $2\,h_0$. This is a clear indication for a strong sensitivity of $E^*(q)$ on the state on swelling and thus, pressure, which needs to be reflected in a faithful model for the contact mechanics, whenever surface ondulations are sufficiently large to induce large interfacial stresses or stress variations.

The effect of solvent quality on the stress profiles at mesoscopic length scales is studied next. Parameters remain similar as before with a focus on the crosslinked ($\rho_{cl} = 0.83$) structure. However, this time, a hydrogel in $\Theta$-solvent ($\chi = 0$) is contrasted with that in a good solvent ($\chi = 0.25$). Results are summarized in Figure 9. Monomers are depleted near the walls for the $\Theta$ solvent compared to good solvent conditions, as shown before in Figure 5a,b for two plane-parallel walls. Moreover, the better solvent increases the osmotic pressure with which the polymers push against the sinusoidal piston. It thus, leads to a larger mean stress but also to a larger oscillatory response in the stress profile, which can be represented quite accurately as a truncated Fourier series via

$$\sigma(x) = \sum_{n=0}^{2} \sigma_n \, \cos(n\,q\,x). \tag{29}$$

If the effective contact modulus $E^*(q)$, i.e.,

$$E^*(q) = 2\,\tilde{\sigma}(q)/\{q\,\tilde{u}(q)\} \tag{30}$$

is expressed in units of the normal stress, the values $E_{dl}^* = E^*(q)/\sigma_0$, at this value of $q$ turn out to be quite close to each other for both considered solvent qualities, i.e., $E_{dl}^* = 0.18$ for the $\Theta$ solvent and $E_{dl}^* = 0.16$ for the good solvent. In the definition of $E_{dl}^*$, stress was defined to be positive when the system is under compression, while $\tilde{\sigma}$ in Equation (30) equals $\sigma_1$ in the Taylor expansion of Equation (29).

However, the $\Theta$-solvent hydrogel reveals noticeably more anharmonicity compared to the one immersed in the good solvent. This can already be perceived with the eye and further substantiated with a qualitative analysis, which shows that the relevance of the first higher-harmonic $\sigma_2$ compared to the ground harmonic $\sigma_1$ is almost twice as large for the $\Theta$ solvent than for the good solvent, i.e., 0.19 versus 0.11. Having a "softer" system shows larger anharmonicity than a more rigid system at identical displacement and wavelength can be seen as counterintuitive from more conventional elasticity.

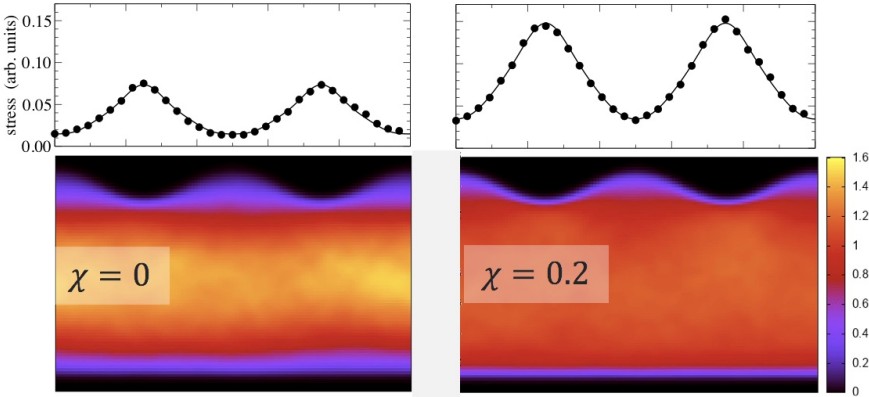

**Figure 9.** Stress profiles (**top**) and monomer concentrations (**bottom**) for a crosslinked hydrogel with a sinusoidal indenter being placed at the original mold height. Θ-solvent (**left**) and good-solvent (**right**) conditions are shown. The full lines in the stress profiles reflect a truncated Fourier series expansion.

### 3.3. Wavelength Dependence of the Contact Modulus

The final result section is concerned with the question at what point $E^*(q)$ can be treated as approximately constant, i.e., above what scale scale-free continuum mechanics can be pursued. Towards this end, results for the dimensionless effective constant $E^*_{\mathrm{dl}}$ were computed as a function of $q$ by proceeding similarly as for the analysis of Figure 9. The such deduced dispersion relations are summarized in Figure 10.

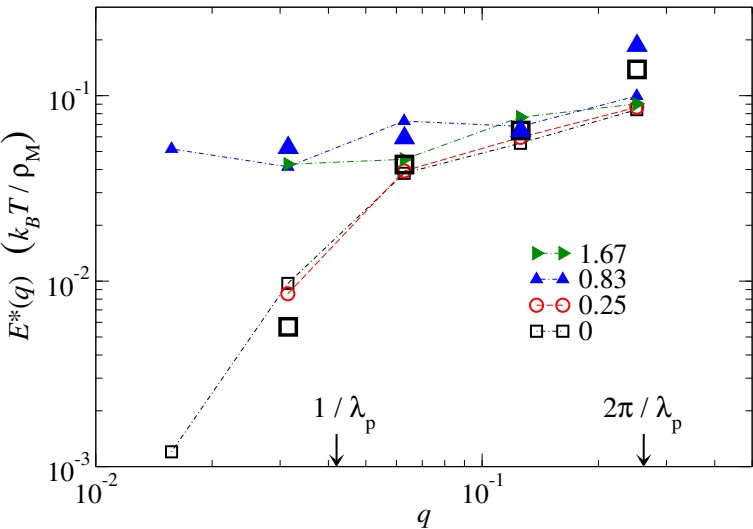

**Figure 10.** Dispersion relation for effective contact modulus $E^*$ in units of thermal energy divided by the monomer density for different concentrations of crosslinkers $\rho_{\mathrm{cg}}$, which state the relative number of crosslink monomers and polymer ends. Connected symbols consider hydrogels, which had been crosslinked in a "hydrophilic" mold attracting the solvent but repelling the polymers, while open symbols show simulation results for hydrogels that were crosslinked in a "hydrophobic" mold leading to a weak adhesion to the polymers. All hydrogels were exposed to the same indenter at a given value $q$ and the mean height was kept constant throughout all simulations. The hydrophobicity of mold and indenter were also kept identical. The default system, shown in Figure 8, corresponds to an undulation with $q = 2\pi/100 \approx 0.063$.

It can be appreciated that for the largest investigated wavevectors, there is little sensitivity of $E^*$ to the degree of crosslinking, unless the crosslinking occurred in a hydrophobic mold, in which case the connected hydrogel has an increased polymer density near its surface than a disconnected polymer solution. The hydrogels appear to show little sensitivity of $E^*(q)$ on $q$ at small $q$, i.e., when

the indenter's wavelengths is less than, say, $\lambda_P/2$, which, in this study, also corresponds roughly to the mean distance between crosslinks. Conversely, the disconnected or partially connected polymer melts show a quickly decaying $E^*(q)$ with decreasing $q$, as to be expected for a melt.

## 4. Discussions and Conclusions

In this work, a coarse-grained model for the simulation of hydrogels was developed with a particular emphasis on its application in the context of contact mechanics. The proposed approach allows one to model not only bulk hydrogels but also to reflect the conditions that existed for the polymers during the crosslinking stage. The arguably strongest aspect of the model is that its description becomes more reliable with increasing degree of polymerization $P$ and increasing distance between crosslinking points, because it is very much designed in the spirit of a self-consistent field theoretical description of polymers [25]. The latter become exact in the limit of large $P$. While revealing many trends correctly, which are summarized further below, the current model certainly needs refinement when applied to specific applications.

First, it needs to be generalized to three dimensions, which, however, is an exercise that can be achieved in a rather straightforward manner. A potential complication is the proper implementation of non-crossing constraints, which were neglected in this 2D study (to avoid artifacts), but which are certainly important to include in 3D. Since two-body repulsion no longer ensures non-crossing at coarse scales, particular measures need to be set into pace [26]. Second, parameters for $k_1$, $k_2$, and $\chi$ must be properly gauged, for example, from measurements of $R_{EE}$ in the bulk and from dilute solutions in the respective solvents, which may differ between production and application. Third, strength and range of adhesion due to the interaction with a confining wall—whether in the mold or with respect to the indenter—need to be parametrized properly and in a size-consistent fashion. This is the aspect, which has probably received the least attention so far in regard to the systematic coarse graining of polymer interactions. Fourth, while our model certainly constitutes a step toward a realistic description compared to existing approaches for constructing hydrogels—such as randomly cutting bonds from an initially crystalline reference structure—there is certainly room for improvement. Rather than imposing crosslinking points to consist predominantly of branching points to three or four chains by design, it would be better to have individual tracers move through the fluid, which could—or perhaps should—be coupled to the (coarse-grained) density of floating polymers. Some realism should then also be added to the geometry of crosslinking points. Last but not least, if the interest extends to dynamical properties, a frequency dependence needs to be added, e.g., with Maxwell models, which are coupled in parallel to the intramolecular springs and/or with descriptions of the solvent's viscosity through the lattice–Boltzmann method [21] or dissipative-particle dynamics [22].

Despite its simplistic nature, our model produces many trends, or at least expectations correctly, which we believe to be important for a qualitative understanding and thus, quantitative reproduction of the contact mechanics of hydrogels. (i) The solvent quality strongly affects the polymer density near the mold walls, even if the bulk concentration is fixed. A stronger depletion of polymers near the walls leads to fewer crosslinks in the hydrogels and thus, to a reduction of elastic moduli (bulk, shear, Young's, and contact modulus) near the walls while that in the bulk is much less affected. This in turn translates into a reduction of the contact modulus at small wavelengths. (ii) The mold's hydrophobicity also affects the polymer structure. A hydrophobic mold surface depletes the solvent (i.e., water) in its vicinity and thereby enhances the monomer density near the surface. This can lead, for example, to a crust-like structure of the resulting hydrogel with extreme stiffening at short wavelengths. (iii) Our model also reproduces a strong dependence of the effectively wavenumber-dependent modulus $E^*(q)$ on the local pressure $p$, i.e., a strong increase of $E^*(q)$ with $p$. This stiffening can be interpreted as being caused by a depletion of (implicit) solvent of a load-bearing hydrogel. (iv) While individual stress profiles in response to sinusoidal indenters reveal non-affine deformations, affinity is reestablished after averaging, at least within—but as we believe probably also beyond—the linear-response regime.

In future models, smaller bond stiffnesses and longer chains between crosslinks will probably be studied to (better) separate the scales and effects due to crosslinking and persistence length.

A systematic comparison of simulations using an atomistic model like ours to continuum approaches appears to be promising. Advanced continuum methods model hydrogels in terms of biphasic poro-(visco)-elastic materials [35–37]. These are top-down approaches: adjustable parameters are gauged on experimental data. When geometries and boundary conditions are changed, parameters can be recycled and predictions be made. However, when the hydrogel architecture is altered, the model has to be reparametrized with experimental input. The approach proposed here is bottom-up: the laws for the statistical mechanics of polymers and solvents at the small scale provide the input and constitutive laws result without further experimental input. This puts tight limits on accessible time- and lengthscales. however, once interactions are properly described, molecular architecture can be changed with a minimum or even no need of further experimental verification. A sensibly defined atomistic/mesoscale approach could also provide a parametrization or even lead to a generalization of currently used constitutive relations. A natural candidate would be the addition of square-gradient corrections to the models with which scale effects are commonly incorporated in continuum theories. Other useful information from our small-scale model to continuum models could be to provide guidelines for how to reflect the non-isotropy of hydrogels in the direction normal to their surfaces.

A comparison of our findings with experiment is difficult, even at a qualitative level. One reason is that experimental studies of the scale-dependence of (visco-) elastic properties in hydrogels are hampered by a scale-dependent time constant. Thus, assessing the quasi-static elastic response requires an adaption of the experimental relaxation rate [38]. Moreover, experiments probing elastic properties at the transition length scale identified in this study are still qualitative in nature [8,39]. Most important, the comparison of our results with experiments is intricate, because indentation experiments are usually done with "single-asperity" indenters, which often can be approximated as Hertzian. Deviations from the expected (Hertzian) contact mechanics, can result from a scale-dependence or from non-linear elasticity or from additional effects. The deconvolution of the various effects, which we achieve by using sinusoidal indenters, appears to be difficult.

The large (local) stress sensitivity of $E^*$ is certainly important to be taken into account in the modeling of the contact mechanics of polymers, because it leads to an important feedback mechanism: If a contact-mechanics description is conducted assuming linear elasticity, some zones in the interface will carry more load than others. The hydrogel stiffens below the high-pressure zones, which means that they carry even more load, which leads to further stiffening. Once pressures are sufficiently large to induce more than a 20% or 50% correction to the original $E^*(q)$, the linear-response assumption can no longer be justified. For randomly rough surfaces, this is expected to happen at the point where the root-mean-square height gradient reaches or surpasses values between 0.2 to 0.5. The subsequent non-linear deformation jeopardizes the efficiency of boundary-value methods at the mesoscale and require major modifications to them to make them reliable down to small scales. In contrast, we foresee that Persson theory [9] can be readily applied to the systems in question but require a reliable determination of $E^*(p,q)$ across the scales for a study along these lines, which is planned for the future.

**Author Contributions:** M.H.M. designed the model, wrote most of the simulation code, conducted the hydrogel simulations, performed their analysis, and wrote most of the manuscript. R.B. proposed the problem and assisted in the write-up of the paper. H.L. assisted in the design of the code and run and analyzed the single-polymer, force-distance simulations.

**Funding:** M.H.M. acknowledges financial support from the DFG through grant Mu-1694/5-2.

**Conflicts of Interest:** The authors declare no conflict of interest.

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
