# Peer review of "Modeling the Contact Mechanics of Hydrogels"

_lubricants, doi:10.3390/lubricants7040035_

Round 1

Reviewer 1 Report

The manuscript under consideration is devoted to molecular dynamics simulations of crosslinked hydrogels. A special form of the interaction potential is introduced, and the scale-dependence of elastic properties has been studied. The paper is interesting and theoretically profound. It is recommended for publication provided the minor revision has been made according to the following remarks.

1. For engineering applications, it would be advantageous to highlight the differences between the predictions of the molecular dynamics (MD) model with continuous mechanics models. For instance, the poroviscoelastic model is widely used for describing the deformational behaviour of hydrogels (see, e.g., Liu, K.; Ovaert, T.C. Poro-viscoelastic constitutive modeling of unconfined creep of hydrogels using finite element analysis with integrated optimization method. Journal of the Mechanical Behavior of Biomedical Materials 2011, 4(3), 440–450. DOI: 10.1016/j.jmbbm.2010.12.005.). The advantage of the linear continuous mechanics models is that they allow analytical treatment (see, e.g., Argatov, I.I.; Mishuris, G.S. An asymptotic model for a thin biphasic poroviscoelastic layer. Quarterly Journal of Mechanics and Applied Mathematics 2015, 68(3), 289–297. DOI: 10.1093/qjmam/hbv008.).

2. Apparently, the presented MD model does not account for any particular mechanism of energy dissipation, and the deformation of hydrogels is assumed to be purely elastic. It is not clear whether the dissipation effect is negligible, or it could be accounted for by some modification of the model.

3. The authors consider the two-dimensional contact periodic contact problem for a hydrogel solid with its upper surface being in frictionless contact with a sinusoidally shaped rigid wall. A similar contact problem for an elastic layer was recently studied with the application to contact deformation of biological tissues (Argatov, I.; Mishuris, G. An asymptotic model for a thin bonded elastic layer coated with an elastic membrane. Applied Mathematical Modelling 2016, 40(4), 2541–2548. DOI: 10.1016/j.apm.2015.09.109.). Considering the referenced paper, it is not clear what boundary conditions are imposed on the bottom surface of the hydrogel layer.

4. Also, from the presented analysis (see Eq.(26)) it is not clear, whether any gradient of the solution in the vertical direction exists. If this gradient is zero, as the right-hand side of Eq.(26) may suggest, the obtained solution is more appropriate for describing the deformation of a relatively thin layer compressed between two wavy rigid walls. 

Author Response

Thank you very much for a very clear and useful report. 

As to point 1: Thank you for providing us with useful references related to the continuum modeling of hydrogels. We actually found a few more and added one of them to the paper. We hope it is OK to have included the discussion in the conclusions section. From a methodological point of view, the main difference between our (generally small-scale models) and the continuum models is that we pursue a bottom-up approach, while continuum mechanics is top-down. Thus, we would never suggest to use our approach to model a macroscopic system. However, we see much potential for it to be used to parameterize, to test, and ultimately to improve the constitutive relations used in continuum models. In fact, we see the door wide open for a beautiful sequential multiscale exercise. And we see potential for our method to explain why small-scale experiments (done with the atomic-force microscope) can lead to unexpected results. 

As to point 2: In the moment, our model refers to thermal equilibrium, which means zero frequency excitations. In this limit, dynamics does not matter – ultimately because the partition function (or free energy) can be decomposed into one purely configurational and one purely momentum-based contribution. We already made a similar, perhaps less clear comment in the first version, but we understand perfectly well that this is easily overlooked in such a long draft. We now state these things explicitly in the introduction, which did not convey an idea about the goals of the original manuscript.  (Referee 2 also rightfully complained about that implicitly.) To fully answer your question: true dynamics can certainly be included. Ways of achieving this would be to use (thermostated) Maxwell elements for bonded interactions plus dissipative particle dynamics, or a coupling to a lattice Boltzmann solver. 

As to point 3: While the eye is usually a very good computer, our systems are effectively semi-infinite. The precise boundary condition at the bottom wall only really matters when the height of an elastic slab is less than, say, the wavelength of the undulation divided by 2 pi.  We added a paragraph at the end of section 2.6, in which we roughly state: The bottom part of the hydrogel is in contact with a frictionless wall, whose z-position is fixed in space. (Thin layers are certainly an interesting topic, but they are not addressed in this work.)

As to point 4, which is very much related to point 3: Your speculation is absolutely justified, but it does not apply. If it did, a clear signal in the E*(q) dependence of the cross-linked network would show up. Specifically, if the system were in the thin-layer limit, E*(q) would increase (exponentially) with decreasing q at small q, because effectively, a zero normal-displacement boundary condition was used at the bottom wall.

Reviewer 2 Report

This is an interesting paper for theoreticians. The authors formulate a general coarse-grained model of a cross-linked polymer, and use molecular dynamics to investigate the effect of microscopic heterogeneity on its elastic properties. However, I find the way the paper is presented confusing, and suggest some changes to its structure.

First, the introduction needs an additional paragraph stating more clearly what the goals of this study are (rather than one only line at the very end).

Second, the paper is somewhat disjointed, with Sections 2.1 - 2.2 disconnected from 2.3 onwards. There is some exploration on the interatomic potential (8), which I find somewhat puzzling since to me it looks like typical harmonic angular and bond potentials. What is exactly the novelty in the proposed potential?

Third, I think the analytical results of the one-dimensional case are interesting and relevant, but I don't see the connection with the rest of the paper. One way to avoid disorientation in the reader is a statement of the kind "The paper is organized as follows..." in the introduction.

Finally, the most relevant results (in my opinion) are those in section 3.2 and 3.3. Can the authors include a plot such as the one in Figure 10, with the x-axis depicting the modulus dependence on the solvent quality parameter? Do the results indicate that a uniform modulus is a good assumption for good quality solvents?

Are the results in Section 3.2 and 3.3 expected to hold in a 3D system?

For the intermolecular interactions, the authors select a potential proportional to the smeared density. Why using this formulation rather than a two-body interaction depending on the distance between the beads? isn't this a straightforward to avoid bond crossing?

Out of curiosity, why did the authors choose to write their own MD code, rather than using an open-source implementation (e.g. lammps) which offers the machinery required for the proposed calculations?

The conclusions should make more emphasis on the key findings, explained in a way that can be understood by theoreticians and experimentalists alike. Here, comparisons with actual materials (even if just to present them as examples of what the model could represent) are needed.

How could these theoretical findings be validated?

"has" instead of "having" in line 79.

"the" instead of "he" in line 143.

"a" instead of"an" in line 181.

"pursued" instead of "pursuied" in line 199.

"densities" instead of "density" in line 215.

"densely" instead of densly" in line 152.

"conventional" instead of "convential" in line 402.

Author Response

Thank you very much for a very clear and useful report. 

As to point “first”: We perfectly understand why we were asked to better reflect the story line in the introduction. In fact, we probably went beyond what was requested and added two quite long paragraphs to the introduction. Quite a few questions could have probably been avoided if we had done that in the first version. 

As to point “second”: The “novelty” is that our bond angle potentials are implicit, while the usually used ones are explicit depending on the bond angle. Actually, we never claimed it to be novel and even clearly conveyed that our model is a small perturbation from those models that we know. We will convey this point even more clearly in the new version of the manuscript. We now also make clear that section 2.1-2.2 are basically reviews of standard polymer physics (with the twist that our bond-angle potentials are implicit and that we remain within a two-dimensional world), while the other sections relate more directly to the hydrogel. 

As to point “third”: The one-dimensional case matters when the persistence length is large. Not having provided the “big picture” and/or the story line in the introduction is probably at the origin of the confusion. Also, the 1D model is the one to be used (even in 2D) when assessing the mean bond length in our result section.  

 As to point “finally”: The paper is already very long and a parameter study could include the variation of many parameters. The next step for us is to extend the model to 3D and to come up with a reasonable parameterization w.r.t. a real system. When we have that in place, we would feel more comfortable to make claims on how solvent quality affect the heterogeneity. But yes, at fixed polymer density, a higher solvent quality should lead to greater homogeneity. 

Other points:

When parameterizing potentials at coarse scales to structural properties, repulsion turns out too soft to avoid bond crossing. This is why bond crossing has to be avoided explicitly in such an approach (see the reference to the Wim Briels’s paper). In addition, as we argued in the manuscript: in 2D, we want to allow bond crossing. Even in 3D, bond crossing can be desirable (before crosslinking) during equilibration. Finally, working with a Flory-Huggins parameter facilitates the parameterization w.r.t. experiment, since FH can be measured almost directly from scattering data. In contrast, two-body repulsion parameters cannot be deduced in simple ways from experiment but requires simulations to be run to fit the parameters.   

It was not clear to us to what extent the model can be realized with LAMMPS, e.g., the coarse-graining or the rather particular interaction of monomers with mathematical walls. Figuring out how to embed these ideas into LAMMPS can take much longer than to write the code from scratch, which took me four days. 

And yes, we expanded the conclusions section, where we discussed why relating our simulations to experiments is tricky at this stage, but also set our model in context with respect to continuum approaches. (Something the other referee asked to be done.)

Thanks for catching typos. We had missed to do a spell check. However, some errors that you caught are not caught by a spell checker. (Perhaps I should stop using vi for LaTeX.) 

Round 2

Reviewer 2 Report

I believe the paper is in good shape to be published, no further revision is required.